# Interaction kinetics between p115-RhoGEF and Gα$_{13}$ are determined by unique molecular interactions affecting agonist sensitivity

Fabian Redlin[1], Anna-Lena Krett[1] & Moritz Bünemann [1✉]

The three RH-RhoGEFs (Guanine nucleotide exchange factors) p115-RhoGEF, LARG (leukemia-associated RhoGEF) and PDZ-RhoGEF link G-protein coupled receptors (GPCRs) with RhoA signaling through activation of Gα$_{12/13}$. In order to find functional differences in signaling between the different RH-RhoGEFs we examined their interaction with Gα$_{13}$ in high spatial and temporal resolution, utilizing a FRET-based single cell assay. We found that p115-RhoGEF interacts significantly shorter with Gα$_{13}$ than LARG and PDZ-RhoGEF, while narrowing the structural basis for these differences down to a single amino acid in the rgRGS domain of p115-RhoGEF. The mutation of this amino acid led to an increased interaction time with Gα$_{13}$ and an enhanced agonist sensitivity, comparable to LARG, while mutating the corresponding amino acid in Gα$_{13}$ the same effect could be achieved. While the rgRGS domains of RH-RhoGEFs showed GAP (GTPase-activating protein) activity towards Gα$_{13}$ in vitro, our approach suggests higher GAP activity of p115-RhoGEF in intact cells.

[1] Institute of Pharmacology and Clinical Pharmacy, Philipps-University Marburg, Karl-von-Frisch-Str. 2, 35043 Marburg, Germany.
✉email: moritz.buenemann@staff.uni-marburg.de

GPCRs represent the largest known group of transmembrane receptors, which act as signal transducers while also displaying a very important drug target. One third of approved drugs target GPCRs[1]. GPCRs can activate G proteins inside the cell. While the heterotrimeric G protein itself consists of an α, a β and a γ subunit, there are four different subfamilies of Gα subunits currently known (Gα$_s$, Gα$_{i/o}$, Gα$_{q/11}$, Gα$_{12/13}$)[2], which lead to distinctly different intercellular responses. The least investigated of these subfamilies, the Gα$_{12/13}$ family, signals through RhoGEFs.

RhoGEFs (guanine nucleotide exchange factors for small GTPases of the Rho family) are known to be key players in the regulation of cell shape, cell differentiation and cell growth[3], through their regulation of the small GTPase RhoA (Ras [Rat sarcoma virus] homolog family member A) and other small GTPases of the Rho family (Cell division control protein 42 homolog [Cdc42] and Ras-related C3 botulinum toxin substrate 1 [Rac1][4]). RhoA can activate more than 60 downstream effectors, many of which have not been investigated properly[5]. One of the effectors, whose effects are relatively well known are Rho kinases (ROCK1/2 [Rho-associated coiled-coil-containing protein kinase ½]), which can phosphorylate more than 15 downstream effectors[6]. This leads to actin cytoskeletal dynamics: F-actin is stabilized through inhibition of Cofilin[7], polymerisation of F-actin is increased and microtubules are stabilized[8]. MRTFA (myocardin related transcription factor A) can be shuttled between cytoplasm and nucleus, as it is regulated by actin cytoskeletal dynamics. Through cytoplasmic G-actin depletion (G-actin bound MRTFA is released) and the generation of F-actin, caused by the activation of the RhoA signaling axis, MRTFA can relocate to the nucleus, where it acts as one of two co-activators for the serum response factor (SRF)[9-11]. Activation of the SRF leads to the activation of immediate early genes and the expression of cytoskeletal proteins that are important for the aforementioned physiological role of this signaling cascade[11].

A subgroup of RhoGEFs called RH-RhoGEFs (RH = RGS [regulators of G protein signaling] homology) can be bound and activated by G proteins of the Gα$_{12/13}$ subfamily and as such act as a link between GPCRs and the RhoA signaling cascade. The three RH-RhoGEFs p115-RhoGEF (p115), LARG and PDZ(postsynaptic density 95, disk large, zona occludens-1)-RhoGEF are similar in their composition, as all of them contain a name-giving RH domain to bind Gα$_{12/13}$ and a PH (pleckstrin homology)/DH (Dbl [diffuse B-cell lymphoma] homology) domain for the binding of effectors. PDZ-RhoGEF and LARG display a 36% and 39% amino acid identity to that of p115-RhoGEF[12]. While activated Gα$_{12/13}$ subunits are able to bind and stimulate the RH-RhoGEFs GEF activity[13], the RH-RhoGEFs themselves exhibit GTPase activating protein (GAP) activity towards Gα$_{12/13}$ in vitro[14], determining the length of interaction between the two proteins. While in vitro GAP activity has been verified for p115-RhoGEF and LARG[15], PDZ-RhoGEF did not show any in vitro GAP activity towards Gα$_{12/13}$[16]. The relevance and balance of this ambivalent role of the RhoGEFs in intact cells has not been clarified and at least for LARG the impact of the GAP activity towards Gα$_{13}$ in intact cells has been questioned, since LARG showed a very long interaction time with Gα$_{13}$ associated with extremely high agonist sensitivity, which is the opposite from what to expect from a relevant GAP activity[17].

Several diseases have been associated with the misregulation of RH-RhoGEFs, such as arterial hypertension[18-21], ischemic heart disease[22], cardiac infarction[23], cardiac hypertrophy[20,24], proliferation defects[25-27], inflammation[28,29] and several forms of cancer[30-40]. Therefore, additional insight into the Gα$_{12/13}$-RH-RhoGEF interaction, which in turn gives more insight into the RhoA signaling cascade as a whole, may prove beneficial to combat the aforementioned diseases.

As RH-RhoGEFs are activated in response to agonists of Gα$_{13}$-coupled GPCRs and in turn may accelerate the deactivation of active Gα$_{13}$ by their GAP activity as demonstrated in vitro[41], their activity and sensitivity to agonists will depend on the lifetime of the Gα$_{13}$ RH-RhoGEF complex. In this study we used a FRET-based approach to compare the interactions of the Gα$_{12/13}$ subfamily with the RH-RhoGEFs p115-RhoGEF, LARG and PDZ-RhoGEF. Using a chimeric approach and site directed point mutations, we identified an interaction motif between the alpha-helical domain of Gα$_{13}$ and a small N-terminal region of p115-RhoGEF to be responsible for fast dissociation kinetics and reduced agonist sensitivity.

## Results

**Interaction of Gα$_{13}$ and p115-RhoGEF visualized by FRET.** To investigate the interaction of Gα$_{13}$ with the RH-RhoGEFs (p115-RhoGEF, LARG, PDZ-RhoGEF) we established a FRET-based assay, where Gα$_{13}$ was labelled with mTurquoise2 between amino acids 127 and 128 and the RH-RhoGEFs were N-terminally tagged with YFP, as described for LARG and Gα$_{13}$[17]. The TP$_α$ receptor (thromboxane receptor) was used, as it is known to activate the Gα$_{12/13}$-family of Gα-subunits[42]. Application of the stable prostaglandin H2 analog U-46619 leads to a conformational change and the activation of the receptor, which is, in turn, transferred to the G protein. The activated Gα-subunits can then interact with the RH-RhoGEFs, thereby recruiting them to the plasma membrane[17,43] to stimulate their GEF activity. The interaction between Gα$_{13}$ and the RH-RhoGEFs can be detected by means of an increase in FRET[17] (Fig. 1a–c), as the interaction partners come into close proximity. We quantified the resulting FRET-signals utilizing the $F_{535nm}/F_{480nm}$ emission ratio that allowed us to monitor single cell signals in high temporal resolution (Fig. 1b). This ratiometric FRET-signal reports the interaction between the two proteins and is used synonymously throughout this work. Whereas YFP-p115 exhibits robust, YFP-LARG moderate and YFP-PDZ-RhoGEF small FRET signals with mTurq2-Gα$_{13}$ (Supplementary Fig. 1a), they all failed to exhibit agonist evoked FRET if coexpressed with mTurq2- Gα$_i$ (Fig. 1d). It was also possible to measure an agonist induced interaction of the three RH-RhoGEFs with Gα$_{12}$ in the same assay, although the complex formation (on-kinetics) seemed to be a lot slower (Supplementary Fig. 2).

**Comparison of Gα$_{13}$-p115-RhoGEF interaction with Gα$_{13}$-LARG interaction regarding agonist sensitivity and off-kinetics.** To compare agonist sensitivity of the interaction between Gα$_{13}$ and the RH-RhoGEFs we determined concentration-response curves by applying different agonist concentrations in a range of 0.05 nM to 30 nM and normalizing the FRET response to 100 nM U-46619 (Fig. 2a). Comparing the resulting curves (fitted by the HillSlope equation, described in the Methods section), the Gα$_{13}$-LARG interaction showed a distinct left-shift compared to the Gα$_{13}$-p115 interaction (Fig. 2b). We previously reported that the slow dissociation between Gα$_{13}$ and LARG was not due to slow agonist dissociation from the TPα-R[17]. As depicted in Fig. 2c the dissociation kinetics of the Gα$_{13}$ interaction with either LARG or p115 after agonist withdrawal did not follow a single exponential decay curve, therefore we quantified dissociation kinetics as the area under the curve (AUC; see Fig. 2c) of the 10 min washout protocol between the two agonist applications that were used to normalize the agonist response to maximum. In this assay, the Gα$_{13}$-LARG interaction showed a significantly higher AUC compared to the Gα$_{13}$-p115 interaction, translating into slower off-kinetics, or longer Gα$_{13}$-RhoGEF interaction time (Fig. 2c).

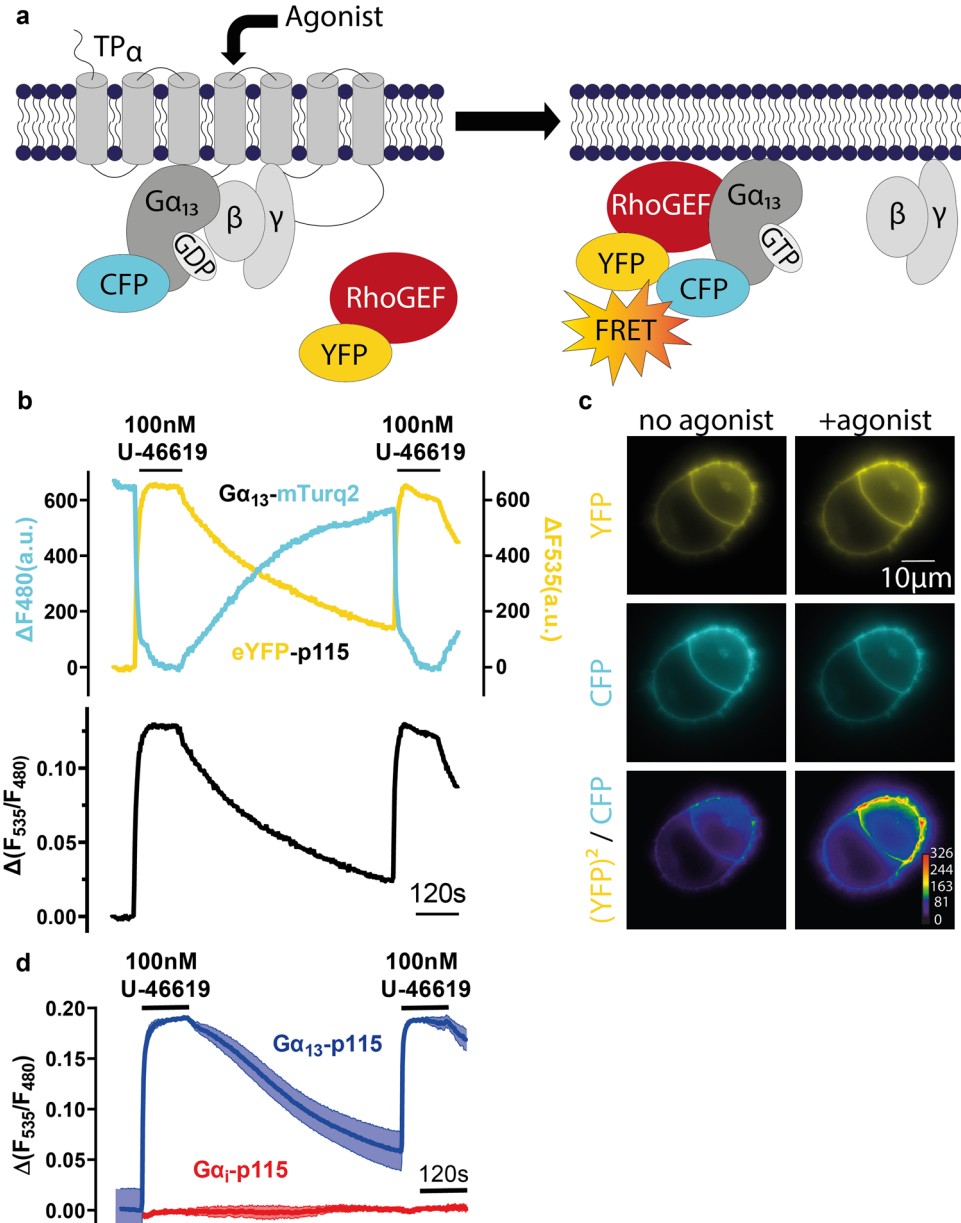

**Fig. 1 Interaction of Gα₁₃ and p115-RhoGEF visualized by FRET. a** To measure the Gα₁₃-p115 FRET-interaction, Gα₁₃ was tagged with mTurqoise2, while p115 was labeled N-terminally with YFP. Upon agonist application the activated GPCR (TPα) activates Gα₁₃, which recruits p115-RhoGEF to the plasma membrane detectable by FRET. **b** Depicted is a representative single cell recording of FRET between Gα₁₃-mTurq2 and YFP-p115. Cells were excited at 425 nm and the emission was simultaneously detected at 480 nm (blue) and 535 nm (yellow). Switching the superfusion buffer to U-46619-containing buffer led to a TPα-mediated interaction of Gα₁₃-mTurq2 and YFP-p115 as reflected by an increase of the $F_{535}/F_{480}$ emission ratio. **c** FRET between Gα₁₃-mTurq2 and YFP-p115 can be observed through agonist application. Multiplication of the YFP image with itself divided by the respective CFP (cyan fluorescent protein) image visualizes the $F_{535}/F_{480}$ emission ratio change (see methods section for image processing). The colors in the (YFP)²/CFP panel represent fluorescence intensity according to the provided scale. **d** Mean±SEM trace of the assay described in A (Gα₁₃-p115 interaction). An emission ratio change can be observed by applying agonist [blue, $n = 8$]. On the other hand, transfection of Gαᵢ-mTurq2 instead of Gα₁₃-mTurq2 did not lead to an emission ratio change [red, $n = 7$], validating the specificity of the assay.

Despite limited FRET amplitudes (Supplementary Fig. 1a) we also measured the Gα₁₃-PDZ-RhoGEF interaction, which displayed an apparently less steep concentration-response curve that was not significantly shifted in relation to Gα₁₃-p115 (Supplementary Fig. 1b). However, we also observed slower on-kinetics after agonist application (Supplementary Fig. 1c). Similarly to LARG, the AUC for the Gα₁₃-PDZ-RhoGEF interaction was also significantly higher compared to p115, even though the agonist sensitivity was comparable (Supplementary Fig. 1c).

**Faster dissociation kinetics of p115-RhoGEF are encoded in N-terminal domain.** To gain insight into why the three RH-RhoGEFs interact differently with Gα₁₃ in terms of binding time and binding sensitivity, we took a closer look at their structural composition. In addition to being shorter, p115-RhoGEF is missing the PDZ domain, when compared to LARG and PDZ-RhoGEF. The PDZ domain allows activation by receptors other than GPCRs[44]. All RH-RhoGEFs possess a DH/PH domain, the binding site for effectors, such as RhoA, as well as the RH

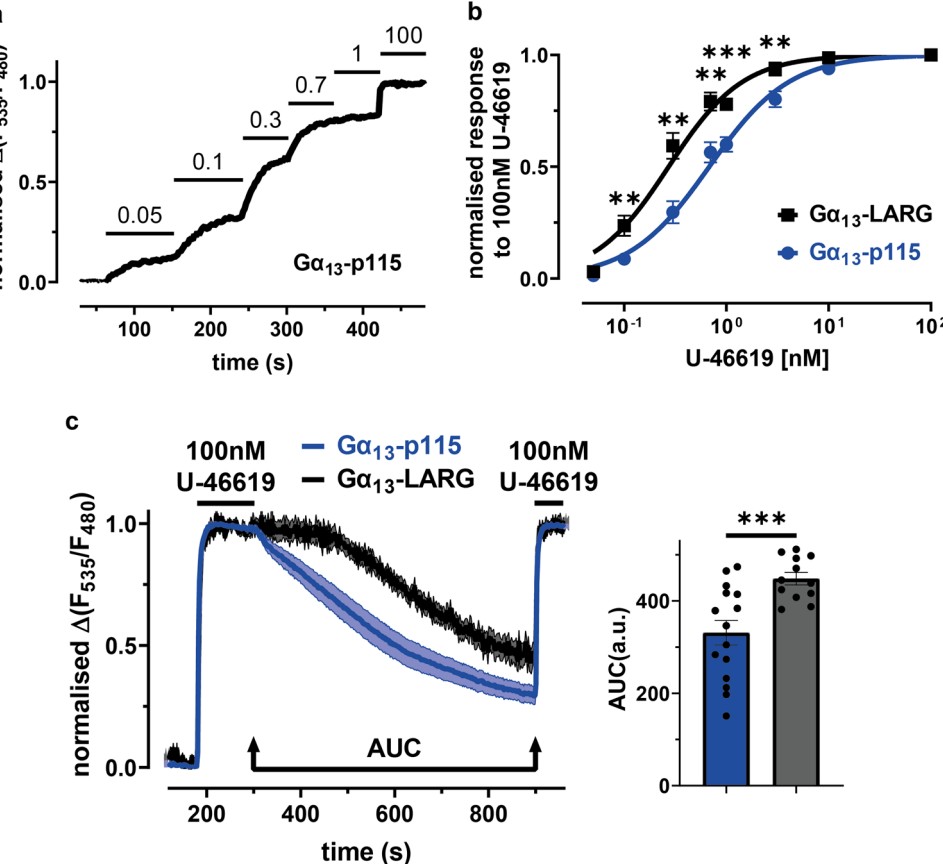

**Fig. 2 Comparison of the Gα₁₃-p115-RhoGEF interaction with Gα₁₃-LARG interaction regarding agonist sensitivity and off-kinetics. a** Illustrated is a representative single cell measurement of the Gα₁₃-p115-RhoGEF interaction. The cell was stimulated with different agonist (U-46619) concentrations to calculate a concentration-response curve. The cells were normalized to the 100 nM U-46619 response. Gα₁₃-LARG interaction data was acquired accordingly. **b** Comparison of the concentration-response curves of the Gα₁₃-p115 [blue] and Gα₁₃-LARG [black, for 0.05 nM, 0.3 nM, 0.7 nM: $n = 8$; for 3 nM, 10 nM: $n = 10$; for 0.1 nM, 1 nM, 100 nM: $n = 18$] interactions. With an $EC_{50}$ value of 0.27 nM the Gα₁₃-LARG curve is left-shifted compared to the Gα₁₃-p115 curve with an $EC_{50}$ value of 0.68 nM. Significant sensitivity differences were found for an agonist range of 0.1 nM to 3 nM [unpaired t-test with Welch´s correction, 0.1 nM: **$P = 0.0076$, 0.3 nM: **$P = 0.0016$, 0.7 nM: **$P = 0.0021$, 1 nM: ***$P = 0.0001$, 3 nM: **$P = 0.0089$]. **c** Mean±SEM trace of the assay described in **1a** for the Gα₁₃-p115 [blue, $n = 15$] and Gα₁₃-LARG [black, $n = 12$] interactions. For quantification, the area under the curve (AUC) of decay between the two agonist applications was compared (marked in the graph). The bar graph shows a highly significant lower AUC for the Gα₁₃-p115 [blue, $n = 15$, mean AUC = 331.1] interaction in comparison to the Gα₁₃-LARG [black, $n = 12$, mean AUC = 448.1] interaction. Statistical analysis was performed using an unpaired t-test with Welch´s correction [***$P = 0.0008$].

domain, the principal binding site for Gα₁₂/₁₃ (Fig. 3a, Supplementary Fig. 1d).

In order to identify the structural differences in the RH-RhoGEFs responsible for the functional differences in the interaction with Gα₁₃, we analysed chimeras between p115-RhoGEF and LARG as well as truncated variants. As the N-terminus of p115 lacks the PDZ domain from LARG and PDZ-RhoGEF (Fig. 3a) we generated chimera 1, containing the N-terminus and RH domain of p115-RhoGEF (amino acids 1-232), as well as the C-terminal segment of LARG (amino acids 559-1544). Chimera1 behaved like full length p115 in the FRET based Gα₁₃-RhoGEF interaction assay with fast off-kinetics (shorter interaction time), contrary the slower off-kinetics of LARG (Fig. 3b). This finding demonstrates a critical role of the N-terminus of p115 regarding the distinct Gα₁₃-RhoGEF interaction kinetics. This result is in line with previous findings that identified the RH domain as the primary Gα₁₃ interaction site[45].

**The region of p115 N-proximal to the start of the RH domain is required for distinct interaction kinetics with Gα₁₃.** To gain further structural insight into the mechanistic basis of the

differential interaction kinetics of LARG and p115 we generated two truncation variants each for p115 and for LARG (Fig. 4a, b), which contained the RH domain including (p115-R, LARG-PR; Fig. 4a) or excluding (p115-Rshort, LARG-Rshort; Fig. 4b) the region N-terminal to the RH domain. The segments C-terminal of the RH domain did not seem to be of importance for the Gα₁₃-p115/LARG interaction kinetics, as the truncations beyond the RH domain (p115-R includes amino acids 1-245 of p115-RhoGEF; LARG-PR includes amino acids 1-558 of LARG) did not show significant differences to their full length counterparts in the interaction kinetics (Fig. 4a). The truncation of the part N-terminal of the RH domain led to a significantly increased AUC of decay in the FRET based interaction assay in the case of p115 compared to full length p115 (p115-Rshort includes amino acids 34-235 of p115; Fig. 4b). In contrast, LARG-Rshort (amino acids 360-561) did not display slowed interaction kinetics with Gα₁₃ compared to full length LARG (Fig. 4b), it rather exhibited a tendency towards faster dissociation kinetics, which did not reach the level of statistical significance. One possibility for the faster Off-kinetics of LARG-Rshort is that the complex of Gα₁₃ with LARG-Rshort becomes more unstable, because of the truncation of the C-terminus, compared to the complex of Gα₁₃ with wt-LARG.

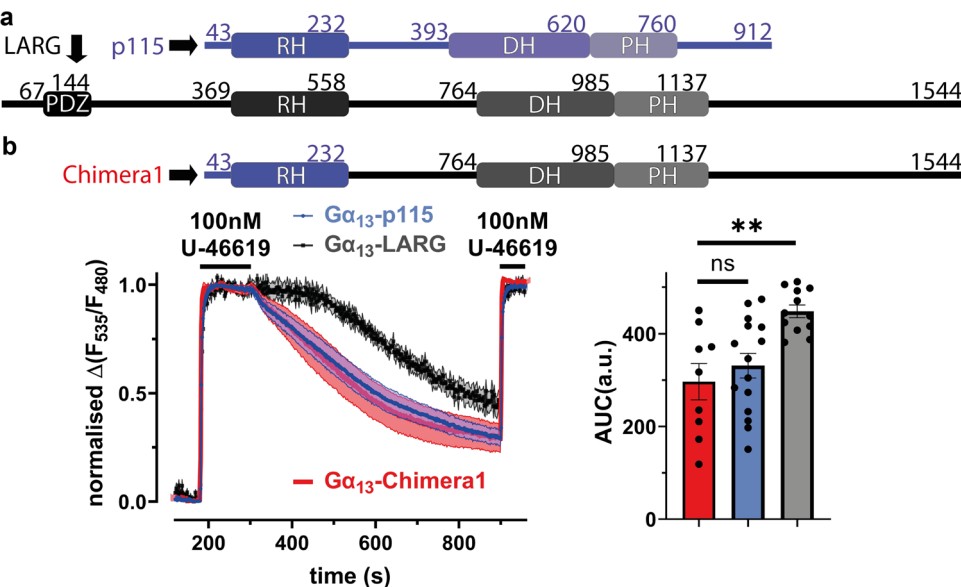

**Fig. 3 Faster dissociation kinetics of p115-RhoGEF are encoded in N-terminal domain. a** Comparing the structures of p115-RhoGEF and LARG: both possess a RH domain, which is the principal binding site for $G\alpha_{13}$, as well as a DH/PH domain for effector binding, while only LARG contains a PDZ domain. **b** Chimera1 includes amino acids 1-232 of p115-RhoGEF (containing the RH domain) in addition to amino acids 559-1544 of LARG. In the previously described FRET-assay, the agonist induced $G\alpha_{13}$-Chimera1 interaction [red, $n = 9$, mean AUC = 269.5] showed a significantly smaller AUC of decay (as described in Fig. 2c) compared to $G\alpha_{13}$-LARG [transparent black, data from Fig. 2c; [**]$P = 0.006$] while showing no significant difference compared to the $G\alpha_{13}$-p115 interaction [transparent blue, data from Fig. 2c, Brown-Forsythe and Welch ANOVA test with Dunnett's T3 multiple comparison test, [ns]$P = 0.7017$].

These results point to a critical role of the short stretch of amino acids proximal to the RH domain, which was previously identified to be important for the GTPase activation activity of the RH-RhoGEFs[45], named rgRGS domain (RhoGEF fragment containing both N-terminal GAP motif and RH domain). In order to test whether differences in the RH domains themselves can be excluded from involvement in the differential interaction kinetics, two Chimeras were created, where the RH domains of p115 and LARG were inserted into full length constructs instead of their native RH domain (Chimera 2: full length p115 (amino acids 1-912) with the RH domain of LARG (amino acids 370-558) instead of the RH domain p115 (amino acids 44-232) and Chimera 3: full length LARG (amino acids 1-1544) with the RH domain of p115 (amino acids 44-232) instead of the RH domain of LARG (amino acids 370-558)). These chimeras proved not to be significantly different to their wild type (wt) counterparts in the $G\alpha_{13}$ interaction kinetics (Fig. 4c), indicating that the RH domains are exchangeable, without affecting major interaction properties of the RH-RhoGEF with $G\alpha_{13}$.

**Identification of molecular interactions between the alpha-helical domain of $G\alpha_{13}$ and p115 required for fast dissociation kinetics.** The complex of a $G\alpha_{13/i}$ chimera and the rgRGS domain of p115 has been resolved by means of x-ray crystallography (PDB ID: 1SHZ)[46,47]. Additionally the complex of native $G\alpha_{13}$ with p115 has also been resolved and the same relevant interactions have been found[48]. In these structure a 12 amino acid motif located proximal to the RH domain of p115 can be found, of which 8 amino acids directly bind to the alpha helical region of $G\alpha_{13}$. The corresponding motif in LARG differs only in two amino acids: aspartic acid at position 356 (instead of glutamic acid at position 29) and glycine at position 359 (instead of glutamic acid at position 32). These two amino acids directly interact with threonine 127 and the basic amino acid arginine 128 of $G\alpha_{13}$ (Fig. 5a, Supplementary Fig. 3).

As the original functional relevant version of the fluorescent $G\alpha_{13}$ was generated by insertion of the fluorophore between these two amino acids, which could potentially interfere with the RH-RhoGEF interaction, we created $G\alpha_{13}(135)$, where the fluorophore mTurquoise2 was placed between the amino acids 135 and 136. The FRET-based interaction assay showed no significant difference in AUC of decay between the two $G\alpha_{13}$ variants with p115 as well as with LARG, proving that the fluorophore in $G\alpha_{13}(127)$ did not impede the interaction with the two RH-RhoGEFs (Supplementary Fig. 4a). Additionally we implemented a BRET-based assay, where in both $G\alpha_{13}$ variants $(127 + 135)$ mTurq2 was replaced for NLuc (Nanoluciferase). We were able to measure full concentration-response curves with both variants in a 96-well plate format utilizing a plate reader, further proving that both variants interact functionally with their effectors (Supplementary Fig. 4b). $G\alpha_{13}(135)$ was used in further experiments to rule out that the interaction was not disturbed in further experiments as well.

In order to test, whether the differences of LARG and p115-RhoGEF within the $G\alpha_{13}$ interaction motif are relevant, we mutated the glutamic acid at position 32 of p115 to the corresponding glycine of LARG (G359) to interrupt the polar interaction with $G\alpha_{13}$-R128. The p115-E32G mutant led to a significantly higher AUC of decay in the FRET based interaction assay, compared to wt-p115, for both $G\alpha_{13}$ variants (Fig. 5b). This means that the interaction time between $G\alpha_{13}$ and the p115-E32G mutant was significantly increased, behaving like the $G\alpha_{13}$-LARG interaction, suggesting glutamic acid on position 32 as the key factor for p115s reduced interaction time with $G\alpha_{13}$. The p115-E32G mutant also showed a left-shifted concentration-response curve (acquisition and evaluation explained in Fig. 2a/b), which translates to higher agonist sensitivity, similar to LARG (Fig. 5c).

Next we checked, if the same result can be achieved by mutating the amino acid in $G\alpha_{13}$ that directly interacts with E32 in p115-RhoGEF, creating the mutant $G\alpha_{13}(135)R128A$. $G\alpha_{13}(135)R128A$, like p115-E32G, showed a significantly higher

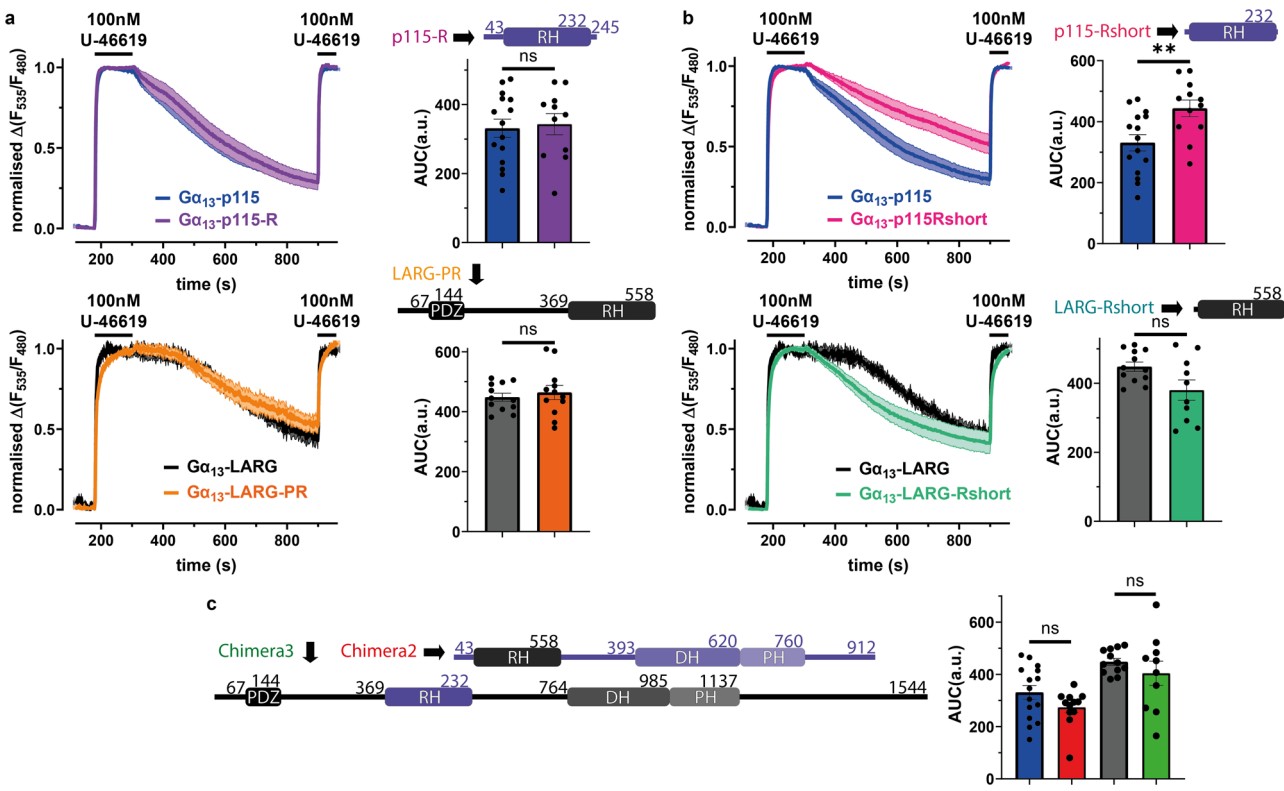

**Fig. 4 The region of p115 N-proximal to the start of the RH domain is required for distinct interaction kinetics with Gα13. a** p115-RhoGEF was truncated at the C-terminal end behind the RH domain to create p115-R (purple, amino acids 1-245, -R means containing the RH domain), while LARG-PR (orange, amino acids 1-558, -PR means containing the PDZ and RH domains) was created the same way. Truncation of p115 [blue, data from Fig. 2c] did not show a significant difference in the AUC of decay (as described in Fig. 2c) [p115-R: purple, $n = 11$, mean AUC = 342.9, unpaired t-test, $^{ns}P = 0.7734$] upon agonist stimulation. The same was true for the truncated version of LARG [black, $n = 12$, data from Fig. 2c vs LARG-PR: orange, $n = 12$, mean AUC = 464.2, unpaired t-test, $^{ns}P = 0.5571$] **b** Additional N-terminal parts were removed to create p115-Rshort (pink, amino acids 34-235) and LARG-Rshort (turquoise, amino acids 360-561). Although LARG-Rshort [turquoise, $n = 10$, mean AUC = 380.2] revealed no significant kinetical difference to wt-LARG [black, data from Fig. 2c, unpaired t-test with Welch´s correction, $^{ns}P = 0.0578$], the AUC of the Gα13-p115-Rshort [pink, $n = 12$, mean AUC = 443.9] interaction was significantly larger than for wt-p115 [blue, data from Fig. 2c, unpaired t-test, $^{**}P = 0.0069$] **c** Chimera 2 [red, $n = 12$, mean AUC = 273.5, unpaired t-test, $^{ns}P = 0.1116$], a full length p115 (amino acids 1-912) containing the RH domain of LARG (amino acids 370-558), did not display a significant difference compared to wt-p115. Chimera 3 [green, $n = 10$, mean AUC = 404.2, unpaired t-test with Welch´s correction, $^{ns}P = 0.3828$], a full length LARG (amino acids 1-1544) containing the RH domain of p115 (amino acids 44-232), did not present a significant difference compared to wt-LARG.

AUC of decay in the FRET based interaction assay. The same was true when both mutants were used simultaneously, proving that the Gα13 R128-p115 E32 interaction plays the key role in reducing Gα13 interaction time with p115 in comparison to the other RH-RhoGEFs (Fig. 5d). As expected the Gα13(135)R128A mutant did not show a significant difference in interaction time with LARG, compared to wt-LARG (Fig. 5d). The same was true for the interaction of the Gα13(135)R128A mutant with PDZ-RhoGEF (Supplementary Fig. 5).

**Insertion of the functional Gα13 interaction motif of p115 into LARG failed to accelerate Gα13-LARG dissociation**. Having identified the interaction of p115 with Gα13 that leads to accelerated dissociation of p115-RhoGEF and Gα13 after agonist withdrawal, we attempted to transfer this interaction to LARG. First we mutated glycine at position 359 to glutamic acid to mimic the binding motif of p115. This mutant (LARG-G359E) did not show a significant difference in AUC of decay in the FRET based Gα13-RhoGEF interaction assay. Neither inserting the complete p115 binding motif (p115 amino acids 20-43 replacing LARG 346-369) into full length LARG (Chimera 4), nor inserting the whole N-terminal part of p115 (p115 amino acids 2-43 replacing LARG 328-369) into full length LARG (Chimera 5), nor replacing the N-terminus of LARG for the N-terminus of

p115 (Chimera 6; p115 amino acids 1-43 replacing LARG 1-369) could shorten the interaction time with Gα13 (Supplementary Fig. 6), suggesting that other yet to be identified structural elements in LARG also play a role in the interaction with Gα13. Although it was not possible to transfer the accelerated dissociation of p115-RhoGEF from Gα13 to LARG, the created chimeras are still functional in their GEF ability towards RhoA. This was tested using a dual-luciferase reporter assay, where the exemplary measurements of cells transfected with Chimera1 (Fig. 3b) and Chimera5 (Supplementary Fig. 6) showed significantly higher relative luciferase activity than cells transfected with an empty pcDNA3 vector, while expectedly cells transfected with p115-R, which does not contain the DH domain (and PH domain) that is required for binding of RhoA and therefore the activation of RhoA, did not show a significant difference (Supplementary Fig. 7).

## Discussion
In order to address functional differences within the family of RH-RhoGEF proteins we studied interactions between Gα13 and p115-RhoGEF, LARG and PDZ-RhoGEF. We originally discovered that LARG has an unusually long interaction time with activated Gα13[17] correlating to a very high agonist sensitivity in intact cells, despite the well described in vitro GAP activity for

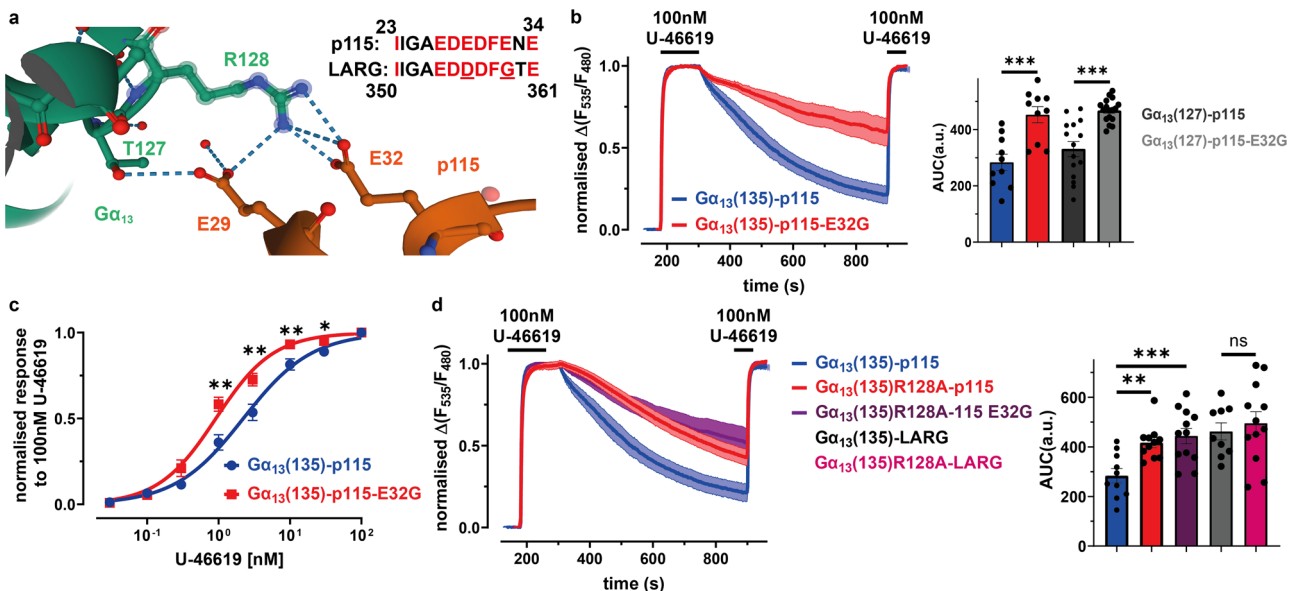

**Fig. 5 Identification of molecular interactions between the alpha-helical domain of Gα$_{13}$ and p115 required for fast dissociation kinetics. a** A crystal structure of the Gα$_{13/i}$-p115-RhoGEF interaction revealed an interaction site located N-terminal of the RH domain of p115 involving E29 and E32 of p115, as well as T127 and R128 of Gα$_{13}$ (Chen, Z., Sprang, S.R. (2005) Crystal Structure of the p115RhoGEF rgRGS Domain in A Complex with Galpha(13):Galpha(i1) Chimera, doi: 10.2210/pdb1shz/pdb)[46,47]. The comparison of this p115 motif with its LARG counterpart exposed a difference of only two amino acids (underlined: E29 and E32 in p115; amino acids in red directly interact with Gα$_{13}$). **b** A single glutamic acid to glycine mutation in the N-terminal binding motif of p115 [red, $n = 10$, mean AUC = 452.6] displayed highly significant slowed off-kinetics compared to wt-p115 [blue, $n = 10$, mean AUC = 283.6, unpaired $t$-test, ***$P = 0.0006$] in the agonist induced interaction (as described in Fig. 2c) with a Gα$_{13}$ where the fluorophore was inserted after amino acids 135. The same was true when a Gα$_{13}$ with the fluorophore inserted after amino acids 127 was used. [p115: dark grey, $n = 15$, mean AUC = 331.1; p115-E32G: light grey, $n = 17$, mean AUC = 467.6, unpaired t-test with Welch´s correction; ***$P = 0.0001$] **c** Comparison of concentration-response curves for the Gα$_{13}$-p115 [blue] and Gα$_{13}$-p115-E32G [red; for 0.03 nM, 0.3 nM, 3 nM, 30 nM: $n = 9$; for 0.1 nM, 1 nM, 10 nM: $n = 10$] interactions. With an EC$_{50}$ value of 0.92 nM the Gα$_{13}$-p115-E32G curve is left-shifted compared to the Gα$_{13}$-p115 curve with an EC$_{50}$ value of 2.22 nM. Significant differences in sensitivity were found for an agonist range of 1 nM to 30 nM [unpaired $t$-test with or without Welch´s correction, dependent on outcome of F-test to compare variances, 1 nM: **$P = 0.0019$, 3 nM: **$P = 0.0059$, 10 nM: **$P = 0.0057$, 30 nM: *$P = 0.0232$]. **d** Mutation of arginine to alanine in position 128, the amino acid that directly interacts with E32 of p115, of Gα$_{13}$ [=Gα13(135)R128A, red, n = 11, mean AUC = 416.9] showed a significantly longer interaction time with p115 compared to wt-Gα$_{13}$ [blue, data from Fig. 5b; **$P = 0.0058$], like the p115-E32G mutation. The same was true when both, the p115 and the Gα$_{13}$ mutant was used [purple, $n = 12$, mean AUC = 443.8; ordinary one-way ANOVA with Tukey´s multiple comparison test; ***$P = 0.0008$]. The Gα$_{13}$(135)R128A-LARG interaction [pink, $n = 12$, mean AUC = 495] however revealed no significant difference compared to the wt-Gα$_{13}$-LARG interaction [grey, $n = 9$, mean AUC = 462.2; unpaired $t$-test, ns$P = 0.7578$].

Gα$_{13}$ of the RH-RhoGEFs p115-RhoGEF and LARG[15,41]. In the present study we found that the Gα$_{13}$-PDZ-RhoGEF complex exhibited comparable dissociation kinetics as LARG, however the interaction of Gα$_{13}$ with p115-RhoGEF was much more short-lived. In addition the Gα$_{13}$-LARG interaction appears to have some kind of lag before decay of the FRET signal. This lag in the mean traces, which is also visible for chimeras 4-6, stems from some cells further increasing Δ(F$_{535}$/F$_{480}$) ratio after agonist withdrawal for several seconds. Why this temporal increase occurs can only be speculated, as we did not find a structural origin. This phenomenon does not appear in every measured cell, which results in the mean trace displaying a plateau before decay of the signal.

By cloning several Chimeras, containing parts of p115-RhoGEF and LARG, we were able to narrow the location of the structural origin of the faster dissociation kinetics of Gα$_{13}$-p115 down to a motif, which directly binds to Gα$_{13}$, located N-terminally of the RH domain of p115. This motif is resolved in a crystal structure of p115 with a chimera of Gα$_i$ and Gα$_{13}$[46] and with native Gα$_{13}$[48] and has been shown to exhibit important interactions with the α-helical domain of the G protein.

Comparing this motif with the corresponding one in LARG (Fig. 5a) and PDZ-RhoGEF (Supplementary Fig. 1e) we found that LARG and PDZ-RhoGEF have nonpolar amino acids (G359 in LARG and P299 in PDZ-RhoGEF) at the position where p115-

RhoGEF has an acidic amino acid (E32). The single amino acid mutation of E32G in p115 led to a key finding of this paper: The interaction time of the p115-E32G mutant with Gα$_{13}$ increased significantly to levels similar as observed for LARG and PDZ-RhoGEF. Importantly, a similar slowing of the dissociation of p115 and Gα$_{13}$ after agonist withdrawal was observed, when the corresponding arginine of Gα$_{13}$ (R128) that engages in the interaction with E32 was mutated to alanine, whereas no additive effect was observed when both mutations were present (p115-E32A and Gα$_{13}$-R128A).

It is rare in biology, that an additional interaction between two proteins accelerate their dissociation. As previously described in detail, the above mentioned motif upstream of the RH domain has been demonstrated to contribute to the GTPase activating properties of the RH-RhoGEF proteins[41,46,49] and was, together with the RH domain, accordingly named rgRGS. This is true for p115-RhoGEF and LARG[15], but not for PDZ-RhoGEF. However it was possible to confer (a less potent) in vitro GAP activity upon PDZ-RhoGEF, by cloning the EDEDF motif of p115 (Fig. 5a), which is one amino acid away from E32, into PDZ-RhoGEF[16]. As E32 of p115 was not inserted into the PDZ-RhoGEF, this residue may not be crucial for in vitro GAP activity, while our results show, that it definitely is for in vivo GAP activity. Nevertheless, the corresponding motif in LARG was also linked to increased GAP activity towards Gα$_{13}$ in vitro[15]. Our experiments actually

do not support the concept, that LARG and PDZ-RhoGEF accelerate deactivation of $G\alpha_{13}$ in intact cells as the dissociation kinetics of the RhoGEF-$G\alpha_{13}$ complex after agonist withdrawal were slow and certainly not faster than $G\alpha_{13}$ $G\beta\gamma$ reassembly in the absence of co-expressed RH-RhoGEFs (Fig. 2c, Supplementary Fig. 1c)[17]. Additionally the RH domains of p115 and LARG without the upstream motif important for GAP activity towards $G\alpha_{13}$ exhibited dissociation kinetics after agonist withdrawal similar to full length LARG or PDZ-RhoGEF, but significantly slower than those observed for p115 (Fig. 4a/b).

No effect of the $G\alpha_{13}$-R128A mutation was observed for the interaction with PDZ-RhoGEF and LARG, further suggesting that in intact cells both RH-RhoGEFs do not lead to detectable GAP activity on the bound $G\alpha_{13}$, whereas in vitro GAP activity of LARG is well described[15]. Currently, the reason for the discrepancy between in vitro data and intact cell data is not known. We failed to transfer faster dissociation kinetics from p115 to LARG by point mutations or even a chimeric approach where the whole rgRGS domain from LARG was exchanged with that of p115, except for one construct, in which the whole N terminal part including the rgRGS domain was replaced by the N-terminus of p115 including the rgRGS domain (Fig. 3b, Supplementary Fig. 6). This suggests, that constraints imposed by some parts of the C-terminus in combination with the RH domain of LARG could prevent the formation of a functional GAP activity. Importantly, the dissociation kinetics of $G\alpha_{13}$-R128A and coexpressed RH-RhoGEFs were similar for all three RhoGEFs, which clearly suggests that the interactions of the N-terminal interaction motif of p115 with $G\alpha_{13}$ are indeed critical for the observed differences in kinetics.

The consequences of a functional interaction of the rgRGS domain of p115 with $G\alpha_{13}$ are not only faster deactivation kinetics but also a significantly lower agonist sensitivity as demonstrated by a left shift of the concentration-response curve of the interaction of $G\alpha_{13}$ with the p115-E32G mutant compared to wt p115 (Fig. 5c). Not only the deactivation kinetics of p115-E32G were similar to those of LARG, also their agonist sensitivity were similar, proving the causative connection between interaction kinetics and agonist sensitivity. Compared to LARG, the concentration-response curve of the $G\alpha_{13}$-PDZ-RhoGEF interaction was right shifted, even though both RH-RhoGEFs exhibited similar dissociation kinetics (Supplementary Fig. 1b/c). As the onset kinetics of the FRET signal between $G\alpha_{13}$ and PDZ-RhoGEF were significantly slower (Supplementary Fig. 1c) we propose that complex formation between PDZ-RhoGEF and $G\alpha_{13}$ is slower, leading to lower agonist sensitivity under steady state conditions compared to LARG. The apparently reduced steepness of the concentration-response curve for the $G\alpha_{13}$ interaction with PDZ-RhoGEF compared to those measured with LARG and p115 should not be overestimated as the signal amplitude in the case of PDZ-RhoGEF was much smaller (Supplementary Fig. 1a) and therefore the signal to noise ratio much reduced, leading to potentially larger errors.

Giving the plethora of diseases the $G\alpha_{13}$-RhoGEF-RhoA signaling axis is involved in, it would be important to understand the functional role of the different RH-RhoGEFs in vivo. As the expression patterns differ for the three different RH-RhoGEFs[12] it is challenging to identify physiological or pathophysiological correlations that originate in functional differences of the three family members. In this context, our point mutants $G\alpha_{13}$-R128A, p115-E32G and the created chimeras that were proven to be functional in their GEF ability (Supplementary Fig. 7), seem to be useful tools to uncover the physiological consequences of the differential G protein interaction kinetics of p115-RhoGEF compared to the other RH-RhoGEFs.

## Materials

**Materials**. Trypsine-EDTA(Ethylenediaminetetraacetic acid), FBS (fetal bovine serum), DMEM (Dulbecco's Modified Eagle's Medium), penicillin/streptomycin and L-glutamine were purchased from Capricorn Scientific, Effectene Transfection Reagent from Qiagen, METAFECTENE PRO from Biontex, NEBuilder® HiFi DNA Assembly Cloning Kit and Q5 Polymerase from New England Biolabs, poly-L-lysine and BSA (bovine serum albumin) from Sigma-Aldrich and U-46619 (9,11-dideoxy-9a,11a-methanoepoxy-prosta-5Z,13E-dien-1-oic acid) from Cayman Chemical.

**Plasmids and agonist**. The following plasmids used in the experiments were: human TPα-R (thromboxane receptor), mouse $G\alpha_{13}$-mTurq2 (monomeric turquoise fluorescent protein 2), eYFP (yellow fluorescent protein)-human LARG, pSRE.L, pRL-TK[17]; human $G\beta_1$-wt, bovine $G\gamma_2$-wt[50]; mouse eYFP-p115[51]; and rat Gαi1[52].

For stimulation of the TPα-R the thromboxane analog U-46619, which was manufactured by Cayman Chemical, Ann Arbor, MI, USA, was used. The preparation of the agonist solutions was performed as previously described[17].

**Cloning and mutagenesis**. The following constructs were cloned using the Gibson Assembly method (a method that does not use restriction sites, instead four primers are used to create an open vector with sticky ends and a fragment with sticky ends that can be ligated). The NEBuilder® HiFi DNA Assembly Cloning Kit has been used according to the manufacturer´s instructions, alongside a Q5 polymerase for PCRs): LARG-PR, Chimera1, Chimera2, Chimera3, $G\alpha_{13}$(135), Chimera4, Chimera5, Chimera6. A list of primers, vector templates and fragment templates used can be found in Supplementary Table 1. The following constructs were cloned using mutagenesis (using a single primer with a difference of few nucleobases in order to exchange a single amino acid): p115-E32G, $G\alpha_{13}$(135)R128A, LARG-G359E (Supplementary Table 1). Additionally, $G\alpha_{12}$-CFP, eYFP-PDZ-RhoGEF, p115-R, p115-Rshort and LARG-Rshort have been created using restriction enzymes (Supplementary Table 1).

**Cell culture and transfections**. Experiments were performed using HEK293T cells (human embryonic kidney cells), which were cultured under standard conditons[53]. For single cell FRET-measurements the cells were transfected according to manufacturer´s instructions with differing plasmids using the Effectene Transfection Reagent (Qiagen, Hilden, Germany), or the METAFECTENE® PRO Transfection Kit (Biontex Laboratories GmbH, Munich, Germany) (used due to limited availability of the Effectene transfection reagent, caused by the COVID pandemic). For the $G\alpha_{13}$-RhoGEF interactions the following amounts of plasmids were used: 0.8 μg TPα-R, 1.0 μg $G\alpha_{13}$-mTurq2, 0.5 μg $G\beta_1$-wt, 0.4 μg $G\gamma_2$-wt and different amounts of RhoGEF constructs, which were chosen subjectively according to their (subjective) expression levels: 1 μg p115-RhoGEF and p115-RhoGEF-based mutants and chimeras, 1.5 μg LARG and PDZ-RhoGEF and mutants and chimeras based on LARG and PDZ-RhoGEF.

**FRET measurements**. Two days prior to the actual FRET measurement, HEK293T cells were transfected as stated above. One day prior to the measurement the transfection was stopped and the cells were split to glass coverslips. The measurements were performed at room temperature, using an inverted microscope (Eclipse Ti, Nikon), equipped with dual excitation and dual emission imaging capabilities[54,55], For the measurements, single cells or small groups of cells were selected based on their fluorescence intensity and distribution. During the

measurement, cells (the donor-fluorophores) were excited with short (60–100 ms) bursts of 425 nm LED-light at 0.5 Hz, while the emission of the acceptor fluorophores were detected simultaneously at 480 (mTurquoise2) or 535 (YFP) respectively and the cells were superfused (pressure-driven; VC3-8xP series; ALA Scientific Instruments) with buffer solution (137 nM NaCl, 5.4 nM KCl, 2 nM $CaCl_2$, 1 nM $MgCl_2$, 10 nM HEPES, 0.1% lipid free bovine serum albumin (BSA) at a pH of 7.3) or buffer solution containing agonist. Data was acquired using the imaging software NIS-Elements (Nikon). As explained in Fig. 1b, the emission intensity of the acceptor fluorophore (YFP) was divided by the emission intensity of the donor fluorophore (mTurquoise2) to create the FRET ratio used to display interactions ($\Delta(F_{535}/F_{480})$). The data was corrected for false excitation and bleed through.

**Dual-Luciferase Reporter Assay System**. The application of the Dual-Luciferase Reporter Assay System by Promega has been described[17]. The assay has been performed in 96-well format in a Tecan Spark 20 M plate reader according to the manufacturer's protocol. HEK293T cells have been transfected using PEI with the following amounts of plasmid for each well: 180 ng pSRE.L, 28.3 ng pRL.TK and 83.3 ng for YFP-p115, YFP-LARG, YFP-p115-R, Chimera1, Chimera5 and pcDNA3. The cells have not been treated with agonist solution.

**BRET measurements**. Some concentration-response curves were acquired utilizing a BRET-based interaction assay in a 96-well plate format using the Tecan Spark 20 M Multimode Microplate Reader. For this purpose mTurq2 in the $G\alpha_{13}(127)$ and $G\alpha_{13}(135)$ constructs were replaced by NLuc. The experimental setup and data evaluation was described previously[56]. Differing agonist concentrations in the first agonist application were always normalized to a second agonist application, containing an agonist concentration that induces a maximum effect.

**Confocal microscopy**. Confocal images, used in Fig. 1c, of either $G\alpha_{13}$-mTorquoise2 or YFP-p115 were acquired using a TCS SP5 (Leica, Wetzlar, Germany) inverted fluorescence microscope[17].

**Statistics and reproducibility**. The $\Delta(F_{535}/F_{480})$ FRET ratio acquired in the FRET measurements was corrected for the peaks, which means that both peaks induced by agonist applications were set to 0 by using the peak analyser (baseline correction) feature of Origin 2018. The rest of the graph was corrected in a linear way according to the fixed peaks. This allowed the comparison of the off-kinetics which were quantified by calculating the area under the curve (AUC) of the decay (see Fig. 2c). In addition the data was normalized (except for Fig. 1d, where absolute amplitudes were compared). Normalization was necessary to acquire standardized results that enabled evaluation in the form of an arbitrary unit (AUC of decay). Normalization was also executed using Origin 2018, graphs were created using GraphPad Prism 8.4. All data is shown in mean ± SEM. Statistical evaluation was performed as stated in the figure legends. Concentration-response curves were fitted with GraphPad Prism 8.4, by using the following equation: Y = Bottom + (Top-Bottom)/(1 + 10^((LogEC50-X)*HillSlope)). The following $p$-values have been used in statistical test for significance throughout the paper: $^{*}P < 0.05$, $^{**}P < 0.01$, $^{***}P < 0.001$

**Reporting summary**. Further information on experimental design is available in the Nature Portfolio Reporting Summary linked to this Article.

**Data availability**

The source data that supports the findings of this study is available from the Supplementary Data.

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

## Author contributions

F. R. and M.B. wrote the manuscript and designed the study and experiments. F.R. and A.K. performed the experiments. M.B. supervised the study.

## Funding

## Competing interests

The authors declare no competing interests.
