## [Peer Review File · Communications Biology]

Reviewers' comments:

Reviewer #1 (Remarks to the Author):

The manuscript by Redlin et al. investigates an important signaling mechanism in cellular physiology and pathology: the down-regulation of a trimeric G protein α subunit ($G\alpha_{13}$) by a specific class of its downstream effectors (RH-RhoGEFs). The study utilizes a clever FRET-based system that this research team has developed previously. Their work identifies a single residue in one RH-RhoGEF, p115, that appears to confer its different GTPase-activating protein (GAP) function to attenuate $G\alpha_{13}$ signaling in comparison to other closely related RhoGEFs. I believe this type of molecular dissection of a G protein-effector interaction is a valuable and sometimes underappreciated approach, and these findings provide a significant advance in the field of G12/13 signaling. I do have some concerns and comments, as follows.

Major Points:

The schematic diagrams (in Fig. 3 and 4, and Suppl. Figures) of all chimeras and truncated forms of RH-RhoGEFs engineered for this study need some clarification. For example, the color scheme of p115 vs. LARG should use colors (or shadings) with better contrast than black vs. dark purple, and the protein regions indicated by thin lines should be made thicker so their colors are apparent. Also, the positioning of labels is not ideal (some next to the polypeptide, some above). The amino acid numbering scheme of the truncations shown in Fig. 4 causes some confusion, as does the change to a different gray/blue-gray shade for the PH domains in Figs. 3 and 4. The text in the Results section supporting/explaining these various constructs needs to be written more clearly. These components of the current study are quite interesting but need to be presented in a way that readers may understand more easily.

I am surprised the authors did not cite nor discuss one of the seminal reports on determinants of GAP activity in the RH-RhoGEFs, by Sprang, Sternweis, and colleagues (Chen et al., 2008, Structure 16: 1532). That paper describes PDZ-RhoGEF as lacking GAP activity toward $G\alpha_{13}$ in vitro, and shows that key amino acid substitutions from p115 can confer GAP activity upon PDZ-RhoGEF. The p115-specific motif substituted into PDZ-RhoGEF by Chen et al. was one residue away from the Glu-32 residue identified in the current manuscript as a key determinant of RhoGEF interaction time with $G\alpha_{13}$. This does not eliminate the novelty of Redlin et al.'s findings, especially in their structural/functional comparison of p115 and LARG, but the juxtaposition with the 2008 results needs to be analyzed and discussed in detail, especially in terms of a mechanism for $G\alpha_{13}$ -specific GAP activity of the RH-RhoGEFs. On a related note, Redlin et al. state multiple times that PDZ-RhoGEF serves as a GAP for $G\alpha_{13}$. The contrary statement and findings about PDZ-RhoGEF in Chen et al. (2008) should at least be addressed in the Discussion.

The Introduction needs more depth in describing the SRF pathway and its relationship to cell shape changes. The actin cytoskeletal dynamics driven by Rho activation are primarily upstream rather than downstream of SRF activation. This section would be strengthened by a brief description of myocardin-related transcription factor A (MRTFA/MAL) as a co-activator of SRF, including how MRTFA trafficking to the nucleus is, in part, governed by cytoplasmic G-actin depletion as F-actin is generated. (See papers on this subject by R. Treisman).

I appreciate that the authors provided data (Suppl. Fig. 3) showing that the change in position of the fluorescent protein tag in $G\alpha_{13}$ (127 vs. 135) did not affect the agonist-induced FRET data for tagged RhoGEFs. However, this evidence would be bolstered by some other type of interaction assay using these differently tagged proteins, such as co-immunoprecipitations.

Minor Points:

The paper that solved the crystal structure of the p115 RH domain in complex with native Ga3 (rather than using a Ga13-Gai1 chimera) should be cited in terms of the contacts between specific residues mapped within these proteins. This paper was Hajicek et al. (2011) J. Biol. Chem. 286: 20625. This is important to include even though the key residue noted in Ga13 (Arg-128) is present in the Ga13-Gai1 chimera.

Check consistency of spelling throughout the manuscript; e.g., signalling vs. signaling, bases vs. basis, trypsin vs. trypsine, principal vs principle. The References section appears to have some misspelled authors. Also, check all Greek symbols throughout the document; I noticed at least one that was missing. Check for erroneous parts of sentences like, "accelerate the turn off the G13 activity.." (p.3). Also, when possessive is given to an object, it reads awkwardly and would benefit from revision. Examples are: "containing p115s N-terminus and RH domain..." (p.5), and "LARGs C-terminal segment" (p.5). The overall manuscript would benefit greatly from a careful round of editing for grammar, usage, and spelling.

All abbreviations need to be accounted for. As one example, "RH" in RH-RhoGEF needs to be defined as rgs homology (and then the term "rgs" needs to be explained and defined also). Terms such as "AUC" need to be defined when they first appear in the document.

The Cloning and Mutagenesis section should be a single paragraph. It is generally best to avoid one- or two-sentence paragraphs, which were noted also in the Results.

Reviewer #2 (Remarks to the Author):

Previous work from this laboratory used an in-cell FRET-based approach to monitor the interactions between the RH-RhoGEFs, LARG and Galpha 13 and showed a surprising long-lived interaction between these proteins after agonist removal. This was particularly interesting given that LARG has GAP activity toward Ga13 in vitro which should lead to rapid Ga13 inactivation and subsequent dissociation. In this manuscript the authors focus on a comparison between LARG and another RH-RhoGEF, p115 RhoGEF. The authors find that the interaction between p115-RhoGEF and Gai13 has a significantly shorter lifetime than that for LARG after removal of agonist. The authors then use a series of chimeras, truncation, and site directed mutants to understand the structural basis for these differences. Interestingly, mutation of a single amino acid in a short region proximal to the RGS subdomain in p115 RhoGEF to the corresponding amino acid in LARG significantly increases the interaction lifetime of p115 to resemble that of LARG. This amino acid is involved in interactions with the alpha helical domain of Ga13 in p115 RhoGEF, and this region of p115 (the N-terminal β N- α N segment of the rgRGS domain) has been previously shown to be necessary and sufficient for GAP activity toward Ga13. Thus, this study suggests that in LARG this interaction has been disabled leading to lower GAP activity and increased complex lifetimes resulting in greater activation potency. These experiments translate previously identified biochemical findings into the context of in cell regulation of these proteins by agonists demonstrating an alteration agonist potency and a likely difference in Rho activation/deactivation kinetics. Overall, the experiments are well done and convincing.

Comments.

1) While generally convincing the decay kinetics seem to have unusual properties that could be explained better. When I look at the comparison between LARG and p115 it appears that there is lag before decay that is longer for LARG than p115, but after that the kinetics look similar. Perhaps the authors have some idea of what is going on here? If not, this should be discussed up front because it is one of the first questions I had.

2) Related to point 1. In later figures when using the chimeras or mutants that conferred "LARG like" decay behavior on p115, the lag is gone and the decay kinetics are simply slower. I feel that this

should be discussed somewhere.

3) on page 8 lines 12 and 13 the authors mention that the proximal N terminal region of the rgRGS domains has been shown to be important for both LARG and p115 GAP activity. They cite a review which when you look at what is referenced in the review, LARG was not examined (J Biol Chem 278: 9912–9919). The authors should cite a primary reference for this and if there is not one, alter this statement.

4) On page 4 lines 5 and 6 the authors state that figure 1 D shows data for PDZ-Rho GEF and LARG which it does not. And data for Gai is also mentioned which is not in Fig 1D. This data is in the supplemental figures.

5) Figure 4B. Why does LARG-Rshort dissociation kinetics get faster if this should be lacking the critical N terminal RGS motif? This does not seem to fit with other aspects of the story where RGS activity is mediating fast dissociation. (Although I will admit the AUC's are not significantly different).

Reviewer #3 (Remarks to the Author):

COMMENTS FOR THE AUTHOR:

In this very interesting manuscript, the authors demonstrated that differences in the interaction of each of the three RH-RhoGEFs (p115-RhoGEF, LARG, and PDZ-RhoGEF) with the Ga13 subunit in high spatial and temporal resolution were examined using a FRET-based single cell assay. In this results, authors found that p115-RhoGEF interacts significantly shorter with Ga13 than LARG and PDZ-RhoGEF, while narrowing the structural basis for these differences down to a single amino acid (p115-E32) in p115-RhoGEFs rgRGS domain using some mutants. This finding is considered to be an extremely interesting discovery in the search for cellular functions of RH-RhoGEFs. However, it is thought that it is necessary to answer the following questions for publication on The Communications Biology.

1. The authors have examined the interaction with Ga13 and show the results of the interaction with Gai in Fig. 1d as the other Ga subunit. There are previous reports that the same subfamily, Ga13 and Ga12, act differently. If the authors have also studied Ga12, it would be necessary to show the results. If they have not, they should use the results of the present study to discuss their thoughts in terms of the structural differences between Ga13 and Ga12.

2. In the present study, the authors are examining the interaction of Ga13 with RH-RhoGEF using various chimeric mutants. How is the activation state of intracellular RhoA altered when these chimeric mutants are used?

If the authors have used FRET techniques or performed pull-down assays with RBDs to examine this, it would be better to show the results. If they have not conducted any studies, it would be better to at least show that any of the chimeric mutants induce RhoA activation.

Reviewer #1:

Major Points:

1) “The schematic diagrams (in Fig. 3 and 4, and Suppl. Figures) of all chimeras and truncated forms of RH-RhoGEFs engineered for this study need some clarification. For example, the color scheme of p115 vs. LARG should use colors (or shadings) with better contrast than black vs. dark purple, and the protein regions indicated by thin lines should be made thicker so their colors are apparent. Also, the positioning of labels is not ideal (some next to the polypeptide, some above). The amino acid numbering scheme of the truncations shown in Fig. 4 causes some confusion, as does the change to a different gray/blue-gray shade for the PH domains in Figs. 3 and 4. The text in the Results section supporting/explaining these various constructs needs to be written more clearly. These components of the current study are quite interesting but need to be presented in a way that readers may understand more easily.”

We thank the reviewer for bringing up this important point. Accordingly, the color scheme of p115 constructs has been changed to lighter blue (from a darker blue), so the difference to the black color scheme of LARG is more apparent. The protein regions indicated by lines have been made considerably thicker. The labels always name the last amino acid of the respective segment. Some labels appeared to be next to the polypeptide. This was the case in the p115-Rshort and LARG-Rshort constructs because there are several amino acids beyond the RH domains in these constructs that were probably hard to see in the schematics. The labels have been removed and the exact amino acid composition of the constructs is instead clarified in the figure descriptions and the results part. In addition the explanations of the constructs has been written more clearly. The shading of the different domains is consistent throughout the manuscript so the domains can be distinguished visually (Figures 3+4, Supplementary Figures 1+6).

(Fig. 3)

(Fig. 4)

(Supplementary Fig. 1)

(Supplementary Fig.6)

2) “I am surprised the authors did not cite nor discuss one of the seminal reports on determinants of GAP activity in the RH-RhoGEFs, by Sprang, Sternweis, and colleagues (Chen et al., 2008, Structure 16: 1532). That paper describes PDZ-RhoGEF as lacking GAP activity toward $G\alpha_{13}$ in vitro, and shows that key amino acid substitutions from p115 can confer GAP activity upon PDZ-RhoGEF. The p115-specific motif substituted into PDZ-RhoGEF by Chen et al. was one residue away from the Glu-32 residue identified in the current manuscript as a key determinant of RhoGEF interaction time with $G\alpha_{13}$. This does not eliminate the novelty of Redlin et al.’s findings, especially in their structural/functional comparison of p115 and LARG, but the juxtaposition with the 2008 results needs to be analyzed and discussed in detail, especially in terms of a mechanism for $G\alpha_{13}$ -specific GAP activity of the RH-RhoGEFs. On a related note, Redlin et al. state multiple times that PDZ-RhoGEF serves as a GAP for $G\alpha_{13}$. The contrary statement and findings about PDZ-RhoGEF in Chen et al. (2008) should at least be addressed in the Discussion.”

The reviewer is completely right. We initially missed to discuss the paper “Recognition of the Activated States of $G\alpha_{13}$ by the rgRGS Domain of PDZRhoGEF” by Chen et al., most likely because our paper was initially focusing on p115-RhoGEF and LARG (at least in terms of experimental work). The paper from Chen et al. gives important context for our work, so we implemented a respective discussion (third paragraph of the introduction: page 4 lines 1-2 and fourth paragraph of the discussion: page 10 lines 20-26). It is of course true that PDZ-RhoGEF did not show GAP activity in vitro. The respective statements have been corrected.

3) “The Introduction needs more depth in describing the SRF pathway and its relationship to cell shape changes. The actin cytoskeletal dynamics driven by Rho activation are primarily upstream rather than downstream of SRF activation. This section would be strengthened by a brief description of myocardin-related transcription factor A (MRTFA/MAL) as a co-activator of SRF, including how MRTFA trafficking to the nucleus is, in part, governed by cytoplasmic G-actin depletion as F-actin is generated. (See papers on this subject by R. Treisman).”

The SRF pathway with regards to MRTFA and actin dynamics has been added to the introduction (paragraph two of the introduction; page 3 lines 16-24).

4) “I appreciate that the authors provided data (Suppl. Fig. 3) showing that the change in position of the fluorescent protein tag in $G\alpha_{13}$ (127 vs. 135) did not affect the agonist-induced FRET data for tagged RhoGEFs. However, this evidence would be bolstered by some other type of interaction assay using these differently tagged proteins, such as co-immunoprecipitations. “

As a second interaction assay we chose a BRET-based approach in a 96-well plate format utilizing a Tecan Spark plate reader. For this purpose we replaced the mTurquoise2 in $G\alpha_{13}$ (127) and $G\alpha_{13}$ (135) for NLuc. Concentration-response curves for both variants were measured with p115-RhoGEF. The results of this independent method confirm the functional interaction of both $G\alpha_{13}$ variants with their effectors. The results of the BRET measurements have been added as Supplementary Figure 4b and the results part has been edited accordingly (subsection five, page 8 lines 9-13).

(Supplementary Fig. 4 a+b)

Minor Points:

5) “The paper that solved the crystal structure of the p115 RH domain in complex with native $G\alpha_3$ (rather than using a $G\alpha_{13}$ - $G\alpha_1$ chimera) should be cited in terms of the contacts between specific residues mapped within these proteins. This paper was Hajicek et al. (2011) J. Biol. Chem. 286: 20625. This is important to include even though the key residue noted in $G\alpha_{13}$ (Arg-128) is present in the $G\alpha_{13}$ - $G\alpha_1$ chimera. “

We thank the reviewer for bringing up this important point. The paper by Hajicek et al. using the native $G\alpha_{13}$ for the crystal structure has been cited as well.

6) "Check consistency of spelling throughout the manuscript; e.g., signalling vs. signaling, bases vs. basis, trypsin vs. trypsin, principal vs principle. The References section appears to have some misspelled authors. Also, check all Greek symbols throughout the document; I noticed at least one that was missing. Check for erroneous parts of sentences like, "accelerate the turn off the G13 activity.." (p.3). Also, when possessive is given to an object, it reads awkwardly and would benefit from revision. Examples are: "containing p115s N-terminus and RH domain..." (p.5), and "LARGs C-terminal segment" (p.5). The overall manuscript would benefit greatly from a careful round of editing for grammar, usage, and spelling."

Spelling has been checked thoroughly and many corrections have been made, especially regarding spelling consistency. Greek symbols have been added where missing, erroneous parts have been corrected and possessives to objects have been removed.

7) "All abbreviations need to be accounted for. As one example, "RH" in RH-RhoGEF needs to be defined as rgs homology (and then the term "rgs" needs to be explained and defined also). Terms such as "AUC" need to be defined when they first appear in the document."

Definitions for all abbreviations have been added in parenthesis upon first use. One exception are words that first appear in the abstract. Because of word restrictions some of these abbreviations cannot be explained in the abstract and are explained on second use instead. This problem could have been solved by a table of abbreviations, which sadly is not allowed in communications biology.

8) "The Cloning and Mutagenesis section should be a single paragraph. It is generally best to avoid one- or two-sentence paragraphs, which were noted also in the Results."

We agree with the reviewer and merged one- or two-sentence paragraphs when possible.

Reviewer #2:

1) "While generally convincing the decay kinetics seem to have unusual properties that could be explained better. When I look at the comparison between LARG and p115 it appears that there is lag before decay that is longer for LARG than p115, but after that the kinetics look similar. Perhaps the authors have some idea of what is going on here? If not, this should be discussed up front because it is one of the first questions I had."

2) "Related to point 1. In later figures when using the chimeras or mutants that conferred "LARG like" decay behavior on p115, the lag is gone and the decay kinetics are simply slower. I feel that this should be discussed somewhere."

We like to thank the reviewer for bringing up these important points. In case of LARG for some of the single cell measurements we observed that the $\Delta(F_{535}/F_{480})$ ratio actually increased after agonist withdrawal instead of decreasing. As this did not happen in every cell (in some the ratio decreased right away) it appears that we have some kind of plateau in the mean trace. Examples of this phenomenon for two single cell G13-LARG measurements:

This temporary $\Delta(F_{535}/F_{480})$ increase does not appear in the G13-p115 interaction or any of the mutants and chimeras based on p115. However the temporary $\Delta(F_{535}/F_{480})$ increase is also observable in Chimeras that are based on LARG. Here are two examples (see Supplementary Fig. 6):

For Chimera4 and Chimera6 (only the AUCs are shown in Supplementary Fig. 6, but not the traces) we can see that this temporal increase of the $\Delta(F_{535}/F_{480})$ ratio is even visible in the mean trace. Why this occurs can only be speculated, but it is most likely connected to the $G\alpha_{13}$ -LARG complex that is forming upon activation. As Chimeras 4-6 (Supplementary Fig. 6) also show this increase one could expect that the part C-terminal of the RH-domain in LARG is responsible for this phenomenon. This on the other hand does not fit with the measurements of $G\alpha_{13}$ -LARG-PR (Fig. 4a) where the C-terminus of LARG was truncated. Here we still have the plateau that does only disappear by additional truncation of the N-terminus (LARG-Rshort, Fig. 4b). In the end we do not get a clear answer from our chimera measurements on why this phenomenon occurs. A short discussion of this issue has been added to the end of the first paragraph in the discussion section (page 9 line 31- page 10 line 4).

3) “on page 8 lines 12 and 13 the authors mention that the proximal N terminal region of the rgRGS domains has been shown to be important for both LARG and p115 GAP activity. They cite a review which when you look at what is referenced in the review, LARG was not examined (J Biol Chem278: 9912–9919). The authors should cite a primary reference for this and if there is not one, alter this statement.”

We thank the reviewer for this correction. Although Chen et al. in J Biol Chem278: 9912–9919 mention GAP activity for p115 specifically in this paper, they also state that “Two members of the p115RhoGEF family, GTRAP48 (13, 15) and PDZRhoGEF,2 bind to $G_{\alpha 13}$ but have little or no GAP activity;”, which we interpret as an implication that LARG has GAP activity. Nonetheless we changed the reference to Suzuki, N. et al. Activation of leukemia-associated RhoGEF by $G\alpha_{13}$ with significant conformational rearrangements in the interface. J. Biol. Chem. 284, 5000–5009 (2009).

4) “On page 4 lines 5 and 6 the authors state that figure 1 D shows data for PDZ-Rho GEF and LARG which it does not. And data for Gai is also mentioned which is not in Fig 1D. This data is in the supplemental figures.”

We apologize for this mistake. We now corrected where to find the data.

5) “Figure 4B. Why does LARG-Rshort dissociation kinetics get faster if this should be lacking the critical N terminal RGS motif? This does not seem to fit with other aspects of the story where RGS activity is mediating fast dissociation. (Although I will admit the AUC’s are not significantly different).”

This finding is actually in line with our results, as we conclude from our measurements, that LARG does not show GAP activity (which is mediated by the RGS motif) towards $G\alpha_{13}$ in vivo. Therefore the removal of the non-functional RGS-motif of LARG did not further slow down the dissociation from $G\alpha_{13}$. It can only be speculated why the Off-Kinetics of LARG-Rshort are faster than wt-LARG (while not significantly). One possibility is that the complex of $G\alpha_{13}$ with LARG-Rshort becomes more unstable, because of the truncation of the C-terminus, compared to the complex of $G\alpha_{13}$ with wt-LARG. A respective statement has been added to the fourth section of the results part: “One possibility for the faster Off-kinetics of LARG-Rshort is that the complex of $G\alpha_{13}$ with LARG-Rshort becomes more unstable, because of the truncation of the C-terminus, compared to the complex of $G\alpha_{13}$ with wt-LARG.”

Reviewer #3:

1. “The authors have examined the interaction with $G\alpha_{13}$ and show the results of the interaction with $G\alpha_i$ in Fig. 1d as the other $G\alpha$ subunit. There are previous reports that the same subfamily, $G\alpha_{13}$ and $G\alpha_{12}$, act differently. If the authors have also studied $G\alpha_{12}$, it would be necessary to show the results. If they have not, they should use the results of the present study to discuss their thoughts in terms of the structural differences between $G\alpha_{13}$ and $G\alpha_{12}$.”

We used the FRET assay described in the paper to take a look at the interaction of $G\alpha_{12}$ with the three RH-RhoGEFs. In our system $G\alpha_{12}$ showed an agonist induced interaction with all three of the RH-RhoGEFs. One thing to notice is that the complex formation (on-kinetics) of G protein and RhoGEF is a lot slower than for the $G\alpha_{12}$ -p115-RhoGEF/LARG interactions. The results of the $G\alpha_{12}$ -RH-RhoGEF interactions in the FRET based assay have been added as Supplementary Fig. 2 and page 5 lines 18-20.

(Supplementary Fig.2)

2. “In the present study, the authors are examining the interaction of $G\alpha_{13}$ with RH-RhoGEF using various chimeric mutants. How is the activation state of intracellular RhoA altered when these chimeric mutants are used?
If the authors have used FRET techniques or performed pull-down assays with RBDs to examine this, it would be better to show the results. If they have not conducted any studies, it would be better to at least show that any of the chimeric mutants induce RhoA activation.”

To ensure that the created chimeras are still functional and can activate their effector RhoA, we utilized the Dual-Luciferase Reporter Assay System by Promega. In previous unpublished work of our group we observed that over-transfecting only Rho-GEFs (e.g. LARG) leads to a substantial activation of the RhoA signaling cascade (rel. luciferase activity), even without stimulation through agonist application. With the previous results in mind we transfected the YFP-p115-RhoGEF and YFP-LARG constructs as positive controls and an empty pcDNA3 vector as negative control. As expected cells transfected with YFP-p115-RhoGEF and YFP-LARG showed significant relative luciferase activity, while cells transfected with the empty pcDNA3 vector did not. As examples for functionality of the chimeric constructs we tested Chimera1 (N-term of p115 and C-term of LARG; see Fig. 3 for structure) and Chimera5 (LARG with inserted binding motif of p115; see Supplementary Fig. 6). Both chimeras showed significant relative luciferase activity and therefore RhoA activation, proving their functionality. As additional prove for the performance of the assay we tested the truncated construct

p115-R for relative luciferase activity. Cells transfected with p115-R did not show significant relative luciferase activity compared to pcDNA3. This is in line with our expectations as p115-R does not contain the DH domain (and PH domain) that is required for binding of RhoA and therefore the activation of RhoA. p115-R not being able to activate the RhoA signaling cascade again proves the functionality of the chimeras created in this work. The results of the Dual-Luciferase Assay System have been added as Supplementary Fig. 7 and explained at the end of the last paragraph in the results section (page 9 lines 12-20).

(Supplementary Fig 7)

REVIEWERS' COMMENTS:

Reviewer #1 (Remarks to the Author):

I am pleased with the authors' detailed, painstaking response to the concerns expressed in my review of their initial submission. It is clear from their rebuttal letter and revised manuscript that they've done an excellent job addressing the major and minor points of my review. I am impressed especially with their improvements to several complex figures (including schematics of mutants), making these easier for readers to interpret. I am comfortable with this manuscript being published in Communications Biology in its current form.

Reviewer #2 (Remarks to the Author):

Thank you for all of the well thought through changes in the manuscript. All of my concerns have been addressed.

Reviewer #3 (Remarks to the Author):

COMMENTS FOR THE AUTHOR:

In this revised manuscript, the authors seem to have responded very carefully and appropriately to the questions and comments of each reviewer. These responses seem to make the manuscript more understandable and interesting to readers. Based on the above, it is thought to be sufficient for publication in Communications Biology.